# Prevalence and Predictive Factors for Exclusive Breastfeeding at Six Months among Thai Adolescent Mothers

**DOI:** 10.3390/children10040682

**Published:** 2023-04-04

**Authors:** Suparp Thaithae, Susanha Yimyam, Pridsadaporn Polprasarn

**Affiliations:** 1Kuakarun Faculty of Nursing, Navamindradhiraj University, 131/5 Khao Road, Wachira Sub-District, Dusit District, Bangkok 10300, Thailand; suparp@nmu.ac.th; 2Faculty of Nursing, Chiang Mai University, 110/406 Inthawaroros Road, Suthep Sub-District, Mueang District, Chiang Mai 50200, Thailand; 3Excellence Center of Community Health Promotion, Walailak University, Nakhon Si Thammarat 80160, Thailand

**Keywords:** exclusive breastfeeding, digital technology literacy, breastfeeding self-efficacy, perceived benefits of breastfeeding, pregnancy intention, family support, occupation, adolescent mothers

## Abstract

Exclusive breastfeeding (EBF) in the first six months of life is the best and the most frequent choice for infants since it has important benefits for the infants and their mothers. However, the exclusive breastfeeding rate in Thailand remains low, especially among adolescent mothers. This predictive correlation study aimed to investigate factors predicting breastfeeding at six months among 253 Thai adolescent mothers from nine hospitals of the Bangkok Metropolitan Administration. Data were collected by using seven questionnaires: the Personal Characteristics, Pregnancy Intention and Breastfeeding Practice, Perceived Benefits of Breastfeeding, Perceived Barriers to Breastfeeding, Breastfeeding Self-Efficacy, Family Support, Maternity Care Practice, and Digital Technology Literacy Questionnaires. Data were analyzed by using descriptive statistics and logistic regression. The findings revealed that the prevalence of EBF at six months among Thai adolescent mothers was only 17.39%, and predictive factors were occupation (work/study) (*p* = 0.034), digital technology literacy (*p* < 0.001), family support (*p* = 0.021), pregnancy intention (*p* = 0.001), breastfeeding self-efficacy (*p* = 0.016), and perceived benefits of breastfeeding (*p* = 0.004). These factors could, together, predict the EBF rate at six months among Thai adolescent mothers in 42.2% (Nagelkerke R^2^ = 0.422) of the cases. These findings may help health professionals to develop activities and strategies for promoting exclusive breastfeeding by increasing breastfeeding self-efficacy, perceived benefits of breastfeeding, and family support, as well as improving digital technology skill among Thai adolescent mothers, especially student/employed adolescents who have unintended pregnancies.

## 1. Introduction

Breast milk provides infants with a variety of nutrients and energy. Breastfeeding helps reduce the morbidity and mortality rates of infants. The World Health Organization (WHO) and the United Nations Children’s Fund [1] recommend and support breastfeeding within the first hour after birth and exclusive breastfeeding during the first six months. Breastfeeding with supplementary food can feed the baby until they are two years old. The WHO aims to increase the rate of exclusive breastfeeding in the first six months to 50% by 2025 and 70% by 2030 [2]. However, the exclusive breastfeeding rate in the first six months is still lower than the target in both developed and developing countries. The global exclusive breastfeeding (EBF) rate at six months was reported to be 44% [3], whereas, in Thailand, the national EBF rate in 2019 was only 14% [1]. In Thailand, 56% of the babies of non-Thai-speaking mothers under six months are exclusively breastfed, while 39% of the babies of Thai-speaking mothers are exclusively breastfed [4], and this is the lowest exclusive breastfeeding rate in Asia [5]. The prevalence of EBF at six months among Thai adolescent mothers in Bangkok was only 19.8% from 2016 to 2017 [6], whilst the national EBF rate was 23.1% in 2016 [7].

Breast milk is best for the infants of adolescent mothers since they often face health problems caused by neonatal complications [8]. Breastfeeding can decrease the severity of the infant’s health problems since breast milk contains antibodies, hormones, and enzymes that reduce infection, inflammation, and neurodevelopment problems [9]. Moreover, breastfeeding could protect against overweight and obesity in later childhood, as well as provide improvements in intelligence [4,10,11,12]. For mothers, producing breast milk can reduce the risk of postpartum hemorrhage, infection, anemia, breast and ovarian carcinoma, weight loss, type 2 diabetes, hypertension, cardiovascular disease, metabolic syndrome, osteoporosis, and rheumatoid arthritis, and they can receive contraceptive benefits following exclusivity [12,13,14]. Breastfeeding helps to establish the all-important and life-long physical and emotional bond between a mother and her baby [4]. Furthermore, breastfeeding reduces health care costs for families and countries [15].

Several studies have found that barriers preventing adolescent mothers from breastfeeding their babies are insufficient breast milk, advice from grandmothers or relatives about providing water to babies, infants crying frequently, and returning to work or study [16]. Thai adolescent mothers may face negative views about inappropriate parenthood, social norms, and family and social pressures and have no bargaining power in their family or the economy [17]. Moreover, the environment of an adolescent mother, including independent ways of life, a lack of social support, social stigma, and difficulty breastfeeding, is an obstacle to breastfeeding [17,18].

As a health promotion, breastfeeding is good for health of infants and mothers which should be promoted as part of daily life considering the conceptual framework from Pender’s Health Promotion Model (HPM) [19]. The HPM has been applied in breastfeeding studies and was found to promote breastfeeding initiation and duration among Latina female mothers in the United States [20].

Breastfeeding can result from personal factors, such as maternal age, marital status, employment, and having an income [6]. At present, digital technology plays an important role and influences daily life, resulting in information that can be found by oneself. Moreover, most teenagers like to use digital technology. When it is not possible to provide face-to-face care, remote provision of breastfeeding education and support could be an important consideration [21]. Therefore, using available information technology, such as computers, mobile phones, tablets, and online media, can provide information for adolescent mothers [22]. Digital technology literacy might be another important factor for promoting breastfeeding among adolescent mothers.

Pregnancy intention is another factor that can predict breastfeeding. Unintended pregnancy causes mothers to develop negative feelings for their infants, which further leads to an inability to create a bond with them. Adolescents feel sadness, depression, and regret from an unintended pregnancy, and some male partners deny the pregnancy; 70–80% of boys disappear from the lives of adolescent girls [23]. This feeling probably influences maternal determination in relation to breastfeeding initiation and duration [24,25].

Previous studies have demonstrated a strong positive association between breastfeeding self-efficacy and EBF in Thai adolescent mothers [26,27]. Based on social cognitive theory [28], perceived self-efficacy is an important motivation that leads the individual to success in their goal. Thus, if adolescent mothers are confident that they are able to breastfeed their babies, they will be patient, not give up, and overcome difficulties to achieve success.

Perceived breastfeeding benefits are a common factor that predicts EBF. In a qualitative study on breastfeeding experiences among Thai adolescent mothers [17], the participants mentioned breastfeeding benefits for their infants and themselves, including how breastfeeding can increase the health of infants and give them a stronger immune system, the advantages of breastfeeding for mothers, the enhancement of maternal bonding, and the cost-saving benefit. Moreover, it revealed that the perceived benefits of breastfeeding were statistically associated with EBF at six months among adolescent mothers in Quito, Ecuador [29], and in Bangkok, Thailand [6].

Adolescent mothers who initiate breastfeeding may require social support from family members, particularly from their partner [17,30,31] and the infant’s grandmothers [18,32,33].

Breastfeeding initiation and continuation can be influenced by the maternity care practices that a mother encounters during her hospital stay. The Ten Steps and the elimination of gift packs containing formula are elements of evidence-based maternity care that can affect breastfeeding. Some adolescent mothers receive maternity care from health care providers that is perceived as positive support. Therefore, maternity care practices may be associated with breastfeeding among adolescent mothers [34].

There is limited knowledge about breastfeeding among Thai adolescent mothers and its predicting factors. There might be differences in the population, lifestyle patterns, and cultures of different countries. In Thailand, maternity leave entitlement for pregnant employees is now set at 98 days per pregnancy (increased from 90 days) and has been extended to include leave taken for pre-natal care, such as to attend medical appointments; the conducive environment for successful six-month exclusive breastfeeding (EBF) needs a significant boost [35]. However, breastfeeding is not a one-woman job, and mothers who choose to breastfeed require encouragement and support from their governments, health systems, workplaces, and families to make it work [4]. Thus, it is necessary to investigate breastfeeding among Thai adolescent mothers to enhance facilitating factors or deal with obstacles that affect exclusive breastfeeding and provide suitable implementation. This study aimed to investigate prevalence and predictive factors for exclusive breastfeeding among Thai adolescent mothers, including maternal age, marital status, education, occupation (work or study), digital technology literacy, pregnant intention, perceived benefits of breastfeeding, perceived barriers to breastfeeding, breastfeeding self-efficacy, family support, and maternity care practice.

### 1.1. Conceptual Framework

Breastfeeding is a behavior that promotes an infant’s health and well-being. In this study, the HPM [19] and relevant literature reviews were used as a conceptual framework since they offer an explanation and prediction of health-enhancing behaviors. This framework consists of three parts: (1) individual characteristics and experiences, including personal factors (such as maternal age, marital status, education, employment, and digital technology literacy) and prior related behavior (such as pregnancy intention); (2) behavior-specific cognitions and affect (such as perceived benefits of action, perceived barriers to action, and perceived self-efficacy), including interpersonal influences (such as family support) and situational influences (maternity care practices); and (3) behavior outcome: exclusive breastfeeding.

Breastfeeding among adolescent mothers relates to cognition and might be influenced by the perceived benefits of and perceived barriers to breastfeeding and breastfeeding self-efficacy. It is the main component of the cognitive and affective domain that is specific to behavior and affects breastfeeding in adolescents. The expected benefits of health-enhancing behaviors have a direct effect on a person’s engagement in such behaviors. Intrinsic benefits can encompass the direct physical impacts of the behavior, while extrinsic benefits can involve monetary or social benefits. Barriers to health-enhancing behaviors influence individuals’ determination to perform such behaviors. Regardless of its level, the readiness to perform a behavior tends to determine the accomplishment of health-promoting activities. Perceived self-efficacy refers to a person’s judgment of their capability to manage and conduct a favorable action. It has a direct effect on triggering health-promoting behavior through an increase in the anticipation of achievement and an indirect effect on the perception of barriers [19]. Furthermore, breastfeeding can result from employment, digital technology literacy, and pregnancy intention as personal factors, family support as an interpersonal influence, and maternity care practices as a situational influence as in Figure 1. 

### 1.2. Key Messages

The low prevalence rate of EBF (only 17.39%) in this present study led to the consideration of the implementation of, for example, peripartum policies and practices for promoting and supporting continued EBF and cooperation with family, school, or workplace for improving EBF rates in adolescents. Remote provision of breastfeeding education and support could be an important consideration when it is not possible to provide face-to-face care, especially during the COVID-19 pandemic.

Breastfeeding self-efficacy, perceived benefits of breastfeeding, and family support, as well as digital technology literacy, should be enhanced sequentially to promote EBF at six months among Thai adolescent mothers, especially student/employed mothers who have unintended pregnancies.

## 2. Methods

### 2.1. Study Design and Setting

This research was cross-sectional and used a predictive correlation design. Thai adolescent mothers were recruited from nine hospitals of the Bangkok Metropolitan Administration when they visited well-baby clinics at six months postpartum.

### 2.2. Participants

Participants were recruited through purposive sampling. Inclusion criteria in-cluded maternal age of 15–19 years, gestational age at birth of 37–42 weeks, no medical compli-cations during pregnancy, labor, or postpartum that interfered with breast-feeding, and no condition in the infant that interfered with breastfeeding. The sample size was calculated using a determination based on the method of logistic regression according to the rule of thumb. If there are predictor variables, there should be a min-imum of 10 sample-size events that result in an event because it gives an estimated re-liable regression coefficient and confidence interval for a rough estimation [36]. In this study, there were 11 primary predictor variables; therefore, the number of samples was 110. This study explored both exclusive and non-exclusive breastfeeding; therefore, the sample size was 2 times that: 220 women. In addition, the researchers increased the sample size by 15 percent to obtain a sample size of 253 women by collecting data from nine hospitals in Bangkok. This study obtained participants that were distributed in proportion to the population of each location and performed simple sampling by drawing with no return, then it received 253 complete questionnaires.

### 2.3. Ethics Consideration

This research was approved by the Human Ethics Committee: Office of the Human Research Ethics Committee Bangkok, Kuakarun Faculty of Nursing and Faculty of Medicine Vajira Hospital, Navamindradhiraj University (E016h/61_EXP, KFN-IRB 2019-18, COA 122/2561). The researcher proposed an explanation of the research objectives and invited volunteers to participate in the study. Informed consent was obtained before collecting data from the participants’ parents in cases involving participants under 18 years of age.

### 2.4. Measurement

Seven questionnaires were used for data collection.

The Personal Characteristics, Pregnancy Intention, and Breastfeeding Practice Questionnaire (PCPiB), which consists of questions on maternal age, marital status, education level, employment, income, pregnancy intention, and breastfeeding practice. For the pregnancy intention question, the responses were assessed on a 10-point scale (1–10), ranging from “unintended” to “intended”. Breastfeeding practices raise concerns about the different types of infant feeding, whether breastfeeding is complete or partial, and whether formula is used.

The Digital Technology Literacy Questionnaire (DTLQ) was developed by Chirasophon and colleagues [22]. It consists of 10 questions about communication tools used, frequency of seeking information, and evaluating media and data when receiving information. Each item is scored on a 5-point Likert scale measured as follows: never = 1, sometimes = 2, often = 3, very often = 4, and always = 5. The items are summed to produce a score ranging from 10 to 50. Higher scores indicate greater digital technology literacy.

The Perceived Benefits of Breastfeeding Questionnaire (PBeBQ) was developed by the researchers based on Pender’s Health Promotion Model. It consists of 30 items based on a literature review and health belief model, including items that cover the benefits of breastfeeding for both infants and mothers. The responses to the questions are categorized as agreement or disagreement. A score of 1 point is given for each correct response. The items are summed to produce a score ranging from 0 to 30. Higher scores indicate a greater perception of breastfeeding’s benefits.

The Perceived Barriers to Breastfeeding Questionnaire (PBaBQ) was developed by the researchers, based on Pender’s Health Promotion Model. It consists of 20 items on breastfeeding barriers, including maternal, infant, and sociocultural barriers. Responses are categorized as agreeing or disagreeing, with 1 point given for agreement. The items are summed to produce a score ranging from 0 to 20. Higher scores indicate a greater perception of breastfeeding barriers.

The Breastfeeding Self-Efficacy Short-Form Scale (BSES-SF) was developed in English by Dennis [37] and translated into a Thai version by Jintrawet and colleagues [38]. It consists of 14 items, each preceded by the phrase “I can always”, and each item is anchored by a 5-point Likert scale, with 1 indicating “not at all confident” and 5 indicating “always confident”. The items are presented positively and summed to produce a score ranging from 14 to 70. Higher scores indicate more confidence to breastfeed.

The Family Support Questionnaire (FSQ) was developed by Biswas [39]. It was translated into Thai by the researchers. It consists of 20 items. Each item is scored on a 5-point Likert scale measured as follows: never = 1, sometimes = 2, often = 3, very often = 4, and always = 5. The total score of family support is computed by summing the scores of each item constructed for the five-item subscale. The total score ranges from 20 to 100, and higher scores indicate more support from family.

The original English version of the Maternity Care Practice Questionnaire (MCPQ) was created by Olaiya and colleagues [34]. It is based on the indicators from the Pregnancy Risk Assessment Monitoring System (PRAMS). The MCPQ was translated into Thai by the researchers. It consists of nine items, eight of which correspond to the Ten Steps for Successful Breastfeeding and one item which assesses the distribution of hospital gift packs containing infant formula. The responses are divided into “yes” and “no”, with 1 point for “yes” and 0 for “no”. The total score is calculated by summing the individual scores. The negatively worded items are reverse scored, and the total score ranges from 0 to 9. Higher scores indicate more maternity care support from the health care system or providers.

All questionnaires were examined for content validity by three experts, including two breastfeeding experts and an information technology expert. A pilot study with 30 postpartum adolescent mothers was conducted. The confidence scores using Cronbach’s alpha coefficient for the DTLQ, PBeBQ, PBaBQ, BSES-SF, FSQ, and MCPQ were 0.83, 0.87, 0.96, 0.98, 0.86, and 0.97, respectively.

### 2.5. Data Collection

The procedure for data collection was carried out by the researchers and research assistants during November 2018 to March 2019. Nine nurses from each of the nine hospitals of the Bangkok Metropolitan Administration were trained as research assistants (RAs) to collect the data and in procedures for human rights protection. The researchers also provided advice to research assistants over the phone and visited each source twice in one month during the five months of the data collection period. The volunteer participants provided informed consent before data collection. It took approximately 25–30 min to complete the questionnaires.

### 2.6. Data Analysis

The data were analyzed using descriptive and inferential statistics. The level of statistical significance was set at 0.05. The personal data of the samples were analyzed with descriptive statistics. The chi-square test was used to compare personal characteristics be-tween exclusively breastfeeding mothers and non-exclusively breastfeeding mothers. The Mann–Whitney U test was used for comparison of selected variables (pregnancy intention, perceived benefits of breastfeeding, perceived barriers to breastfeeding, breastfeeding self-efficacy, family support, digital technology literacy, and perceived maternity care practice) between exclusively breastfeeding mothers and non-exclusively breastfeeding mothers. Logistic regression statistics were used to test the prediction ability of variables in relation to exclusive breastfeeding among Thai adolescent mothers.

## 3. Results

### 3.1. Prevalence of Exclusive Breastfeeding among Adolescent Mothers

Of the 253 adolescent mothers in this study, there were 44 (17.39%) with exclusive breastfeeding at six months postpartum, whereas non-exclusive breastfeeding (n = 209, 82.61%) included a high number of 140 mothers (66.99%) who stopped breastfeeding, and 69 mothers (33.01%) fed their babies breastmilk combined with formula (Table 1). There was no statistically significant difference for personal characteristics (age, marital status, education level and occupation) between exclusive breastfed mothers and non-exclusive breastfed mothers (Table 2). Digital technology literacy, pregnancy in-tention, perceived benefits of breastfeeding, breastfeeding self-efficacy and family support were statistically significant difference. Additionally, perceived barriers to breastfeeding and perceived maternity care practice were no statistically significant difference (Table 3).

### 3.2. Predictive Factors for Exclusive Breastfeeding among Thai Adolescent Mothers

The selected factors for EBF at six months, including maternal age group, marital status, education level, occupation (work/study), digital technology literacy, pregnant intention, perceived benefits of breastfeeding, perceived barriers to breastfeeding, breastfeeding self-efficacy, family support, and maternity care practice, were investigated as the predictors. As shown in Table 4, the logistic regression analysis revealed that the significant predictors of exclusive breastfeeding at six months postpartum were occupation (*p* = 0.034; OR 2.73, 95% CI 1.08–6.89), digital technology literacy (*p* < 0.001; OR 1.24, 95% CI 1.11–1.39), family support (*p* = 0.021; OR 1.38, 95% CI 1.17–2.31), pregnancy intention (*p* = 0.001; OR 1.24, 95% CI 1.09–1.41), breastfeeding self-efficacy (*p* = 0.016; OR 1.05, 95% CI 1.01–1.10), and perceived benefits of breastfeeding (*p* = 0.004; OR 1.79, 95% CI 1.67–1.93). These predictive factors could explain 42.2% (Nagelkerke R^2^ = 0.422; log likelihood = 149.794) of the variation in breastfeeding among the participants. Housewives and unemployed adolescent mothers showed 2.73 times higher EBF at six months than student or employed adolescents (*p* = 0.034; OR 2.73, 95% CI 1.08–6.89). Adolescent mothers with high digital technology knowledge were 1.24 times more likely to have EBF at six months compared to adolescent mothers with low digital technology literacy (*p* < 0.001; OR 1.24, 95% CI 1.11–1.39). Adolescent mothers with high family support were 1.38 times more likely to have an EBF at six months compared to adolescent mothers with low family support (*p* = 0.021; OR 1.38, 95% CI 1.17–2.31). Adolescent mothers with high pregnancy intentions were 1.24 times more likely to have an EBF at six months compared to adolescent mothers with low pregnancy intentions (*p* = 0.001; OR 1.24, 95% CI 1.09–1.41). Adolescent mothers with high breastfeeding self-efficacy were 1.05 times more likely to have six months of EBF compared to adolescent mothers with low breastfeeding self-efficacy (*p* = 0.016; OR 1.05, 95% CI 1.01–1.10). Adolescent mothers with high perceived benefits of breastfeeding were 1.79 times more likely to EBF at six months compared to adolescent mothers with low perceived benefits of breastfeeding (*p* = 0.004; OR 1.79, 95% CI 1.67–1.93).

## 4. Discussion

In this study, the prevalence of EBF among Thai adolescent mothers at six months was only 17.39%. This rate is lower than has previously been reported at 19.8% in Bangkok between 2016 and 2017 [6] and in Northern Thailand (36.4%) in 2017 and 2018 [27]. Moreover, the target rate of the 12th Na-tional Economic and Social Development Plan (2017–2021) for six-month-old infants should be at least 30%. This low prevalence rate of EBF in the present study leads to the consideration of strategies or their implementation for improving EBF rates in adolescents, such as peripartum policies and practices to promote and support EBF, cooperation with family, school, or the workplace. Moreover, not only maternal education in clinics but also digital technology information should be provided for adolescent mothers to encourage continued breastfeeding. Remote provision of breastfeeding education and support (such as video chat, wechat, facebook, social media, and telephone-based interventions, etc.) could be an important consideration when it is not possible to provide face-to-face care, especially during the COVID-19 pandemic [21].

### 4.1. Predictive Factors of EBF

This study found that significant predictive factors of EBF at six months among adolescent mothers are employment, pregnant intention, digital technology literacy, family support, breastfeeding self-efficacy, and perceived benefits of breastfeeding (Table 4). This finding could be explained as follows.

### 4.2. Occupation (Work/Study)

Unemployed mothers or housewives have a 2.73 times greater rate of EBF than employed mothers and students since breastfeeding, combined with returning to studies with increased fatigue due to a sleep deficit and milk leakage, affects the self-images of the latter. In addition, maternity leave policies for students and day care centers for adolescent mothers are not available in Thai schools. This finding is consistent with other Thai studies [6,16,31], which found that the most common reason that causes mothers to discontinue EBF is the obstacle of an occupation. Furthermore, the necessity of returning to work or school was the most common reason for discontinuing EBF before six months in adolescent mothers [6].

### 4.3. Digital Technology Literacy (DTL)

The findings found that digital technology literacy seems to be an important factor predicting EBF at six months among adolescent mothers. Adolescent mothers with high digital technology literacy were 1.24 times more likely to be EBF at six months compared to adolescent mothers with low digital technology literacy (Table 4, *p* < 0.001). It could explain why there were more chances for adolescent mothers with high digital tech-nology literacy to exclusively breastfeed than those with low digital technology literacy. Since most postpartum adolescent mothers searched the internet for information when having problems breastfeeding. Digital technology tools and resources are used to capture, handle, store, and exchange information via electronic communication. The acquisition of knowledge from the internet was a way to increase knowledge and skills in breastfeeding. This finding is supported by other systematic review studies, found that remotely provided breastfeeding support significantly reduced the risk of women stopping exclusive breastfeeding at six months by 15% [21]. Furthermore, it revealed that combining educational activities with web-based, personalized support through discussion forums appears to be the most effective way to improve breastfeeding outcomes and long-term exclusive breastfeeding rates [40].

### 4.4. Family Support

Family support, as an interpersonal influence of the HPM, may affect an adolescent’s breastfeeding directly or indirectly through social pressures or motivation to commit to a plan of action [41]. In consistency with other studies, breastfeeding support relies on family members who have had positive breastfeeding experiences, particularly the partners [17,30,31] and infants’ grandmothers [18,32,33]. Family support provides information on the benefits of breastfeeding for infant health, as well as provides role models in a breastfeeding environment, which influences breastfeeding initiation, while the quality of support and assistance in overcoming breastfeeding difficulties increases breastfeeding duration [42].

### 4.5. Pregnancy Intention

In this study, pregnancy intention could predict EBF in the first six months among Thai adolescent mothers. This can be explained by the fact that the pregnancy intention motivates them to continue exclusive breastfeeding. Adolescent mothers who are determined to become pregnant and play a maternal role in order to have children assume the role of a mother, including breastfeeding. Even when faced with various problems, they endure them and try to find solutions to deal with them until they are able to breastfeed their babies. They try to learn about breastfeeding knowledge from nurse–midwives and search for information from other sources using digital technology. This finding is consistent with other studies [24,25], which showed that pregnancy intention is predictive of breastfeeding duration.

### 4.6. Breastfeeding Self-Efficacy

Breastfeeding self-efficacy is another influential predictor for EBF at six months. In this study, adolescent mothers with high breastfeeding self-efficacy were more likely to be practicing EBF for six months than those with low breastfeeding self-efficacy (Table 4, *p* = 0.016). A possible explanation could be how the adolescent mothers in this study perceive their ability or confidence to breastfeed their infants. As the HPM hypothesizes that perceived self-efficacy in a given behavior increases, the likelihood that a given behavior will continue also increases [19]. Breastfeeding self-efficacy is the main component of the cognitive and affective domain in the Pender model that is specific to behavior and affects the breastfeeding of adolescents. Self-efficacy is the best predictor of health promoting behaviors in adolescents for both normal health and illness conditions [26]. The finding in this study is congruent with previous studies showing that breastfeeding self-efficacy influences breastfeeding outcome [26,27].

### 4.7. Perceived Benefits of Breastfeeding

As in other studies [6,17,29,30], in this study, adolescent mothers with high perceived breastfeeding benefits were more likely to practice EBF for six months than those with low perceived breastfeeding benefits (Table 4, *p* = 0.004). Most commonly, mothers reported perceiving infant health benefits in providing the best possible nutrition. Adolescent mothers continue breastfeeding when they perceive that there are benefits related to it. Since then, this belief has been linked to the mothers’ motivation to persist in their breastfeeding.

## 5. Conclusions and Implementation

The results of this study revealed that occupation (work or study), digital technology literacy, family support, pregnancy intention, breastfeeding self-efficacy, and perceived breastfeeding benefits can predict breastfeeding among Thai adolescent mothers.

Nurse–midwives and other health professionals should assess pregnancy intentions. The partners and husbands of adolescent mothers, and the infants’ grandmothers, should also be included in the activities or intervention. There should be coordination in promoting breastfeeding between hospital and community nurses, educational institutions, and enterprises and in facilitating the breastfeeding of adolescent mothers by providing favorable places and allocating time to promote breastfeeding with mothers of other ages. Moreover, understanding unique adolescents and their needs should be emphasized in breastfeeding promotion programs via maternal education in health facilities and digital technology information in order to enhance breastfeeding self-efficacy and the perceived benefits of breastfeeding and increase exclusive breastfeeding. Strategies and activities to overcome breastfeeding barriers within schools, workplaces, and communities should be supported and encouraged among adolescent mothers and their families.

## Figures and Tables

**Figure 1 children-10-00682-f001:**
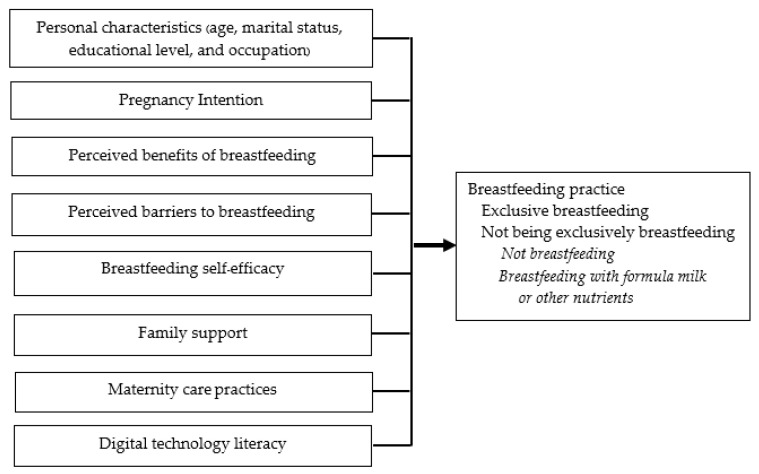
Research conceptual framework.

**Table 1 children-10-00682-t001:** Breastfeeding practice of adolescent mothers (*n* = 253).

Breastfeeding at Six Months	Frequency	Percentage
Exclusive breastfeeding	44	17.39
Non-exclusive breastfeeding	209	82.61
Breastfeeding with formula	69	33.01
Not breastfeeding (formula)	140	66.99

**Table 2 children-10-00682-t002:** Comparison of personal characteristics between exclusively breastfeeding mothers (*n* = 44) and non-exclusively breastfeeding mothers (*n* = 209).

Personal Characteristics	Total (*n* = 253)	EBF (*n* = 44)	Non-EBF (*n* = 209)	*p*-Value
**Age** (**years**)	17.81 (1.28)	17.73 (1.37)	17.83 (1.27)	0.622
13–16	32 (12.65%)	7 (15.91%)	25 (11.96%)	
17–19	221 (87.35%)	37 (84.09%)	184 (88.04%)	
**Marital status**				0.535
Separated/divorced	49 (19.37%)	10 (22.73%)	39 (18.66%)	
Married	204 (80.63%)	34 (77.27%)	170 (81.34%)	
**Education**				0.694
Primary school	13 (5.14%)	3 (6.82%)	10 (4.78%)	
Senior high school	134 (52.96%)	21 (47.73%)	113 (54.07%)	
Junior high school/Vocational school	106 (41.90%)	20 (45.45%)	86 (41.15%)	
**Occupation** (**work/study**)				0.011
Employed/studying	202 (79.84%)	29 (65.91%)	173 (82.78%)	
Unmployed/housewife	51 (20.16%)	15 (34.09%)	36 (17.22%)	

**Table 3 children-10-00682-t003:** Comparison of selected variables between exclusively breastfeeding mothers (*n* = 44) and non-exclusively breastfeeding mothers (*n* = 209).

Selected Variables	Total	EBF (*n* = 44)	Non-EBF (*n* = 209)	*p*-Value
	Mean (SD)	Mean (SD)	Mean (SD)	
Digital technology literacy	11.63 (5.79)	16.00 (3.01)	10.74 (5.82)	<0.001
Pregnancy intention	4.39 (3.63)	6.07 (3.89)	4.04 (3.48)	0.001
Perceived benefits of breastfeeding	24.96 (5.03)	26.05 (3.72)	24.74 (5.24)	<0.001
Perceived barriers to breastfeeding	4.33 (4.06)	3.10 (2.03)	4.78 (4.21)	0.322
Breastfeeding self-efficacy	52.23 (13.68)	60.64 (13.16)	50.52 (13.16)	<0.001
Family support	76.64 (17.20)	82.26 (17.33)	75.50 (16.99)	0.011
Perceived maternity care practice	8.81 (1.01)	9.05 (0.31)	8.76 (1.09)	0.126

**Table 4 children-10-00682-t004:** Logistic efficiency of prediction variable for EBF among Thai adolescent mothers (*n* = 253).

Variables	Odds Ratio	B	S.E.	Wald	95% CI		*p*-Value
					Lower	Upper	
**Age** (**years**)							
13–16 vs. 17–19	1.65	0.50	0.81	0.37	0.33	8.11	0.541
**Marital status**							
Separated/divorced vs. married	1.32	0.28	0.55	0.25	0.45	3.89	0.616
**Education**							
Primary school	Reference						
Senior high school	2.88	1.06	1.29	0.67	0.23	36.01	0.413
Junior high school/vocational school	1.13	0.12	0.47	0.07	0.45	2.85	0.799
**Occupation**							
Employed vs. unemployed	2.73	1.00	0.47	4.50	1.08	6.89	0.034
**Digital technology literacy**	1.24	0.22	0.06	14.53	1.11	1.39	<0.001
**Family support**	1.38	0.21	0.03	5.60	1.17	2.31	0.021
**Pregnancy intention**	1.24	0.22	0.07	10.78	1.09	1.41	0.001
**Breastfeeding self** **-efficacy**	1.05	0.05	0.02	5.78	1.01	1.10	0.016
**Perceived benefits of breastfeeding**	1.79	0.24	0.08	8.50	1.67	1.93	0.004
**Perceived barriers to breastfeeding**	0.96	−0.04	0.05	0.64	0.88	1.06	0.424
**Perceived maternity care practice**	1.96	0.67	0.60	1.25	0.60	6.35	0.263

Log likelihood (149.794), Nagelkerke R^2^ (0.422).

## Data Availability

Not applicable.

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
