# Peer review of "Prevalence and Predictive Factors for Exclusive Breastfeeding at Six Months among Thai Adolescent Mothers"

_children, 2023, doi:10.3390/children10040682_

Round 1

Reviewer 1 Report

The theme is very interesting.

The study is well articulated and since it addresses modifiable aspects of the approach to improving breastfeeding it becomes more significant for the achievement of objectives. Then, it affects a category particularly at risk of non-breastfeeding and therefore the care towards this type of mother must be intensified and better structured.

The number of adolescent mothers is missing. It would be useful to know the incidence of adolescent pregnancies in the city and state to which the study refers to give the size of the problem and therefore the importance of the study.

For adolescent mothers, breastfeeding can reduce the risk of postpartum hemorrhage, breast and ovarian carcinoma, and lower the risk of type 2 diabetes (Victora et al., 2016). Furthermore, breastfeeding reduces healthcare costs for families and countries (WHO & UNICEF, 2020).

I would remove for adolescent mothers because it is valid for all mothers

I would enrich the bibliography on the advantages of breastfeeding.

“This study aimed to investigate prev-alence and predictive factors for exclusive breastfeeding among Thai adolescent mothers, including maternal age, marital status, education, occupation (work or study), digital tech-nology literacy, pregnant intention, perceived benefits of breastfeeding, perceived barri-ers to breastfeeding, breastfeeding self-efficacy, family support, and maternity care prac-tice.

This list is repeated too many times

“The Perceived Benefits of Breastfeeding Questionnaire (PBeBQ) was developed by the researchers through literature review and theoretical assertion of Pender’s Health Promo-tion Model”

Repetition of this phrase should be avoided

“Adolescent mothers who are determined to become pregnant and play a maternal role in order to have children will assume the role of a mother, including breastfeeding”

How many adolescent mothers have intentionally become pregnant? It should be specified.

The description of the calculation of the various scores should be simplified because it is difficult to understand.

The conclusion should be more succinct.

Author Response

Dear  Reviewer

The researcher has already made corrections according to the suggestions.

Please take it into consideration. by using red letters to show the affected area.

Kind regards,

Pridsadaporn Polprasarn

Reviewer 2 Report

The work is interesting because it provides data on breastfeeding of adolescent girls from an environment far away from us. Some circumstances need to be explained in more detail, which is indicated in the text. The advantages of breastfeeding should not be talked about as a potential possibility, but as health-proven facts.

Author Response

(The authors gave the same response as above.)
